# CNN_FunBar: Advanced Learning Technique for Fungi ITS Region Classification

**DOI:** 10.3390/genes14030634

**Published:** 2023-03-03

**Authors:** Ritwika Das, Anil Rai, Dwijesh Chandra Mishra

**Affiliations:** Division of Agricultural Bioinformatics, ICAR-Indian Agricultural Statistics Research Institute, New Delhi 110012, India

**Keywords:** CNN, fungi ITS, *k*-mer, KNN, Naïve-Bayes, random forest, SVM, taxonomy, topsis, UNITE

## Abstract

Fungal species identification from metagenomic data is a highly challenging task. Internal Transcribed Spacer (ITS) region is a potential DNA marker for fungi taxonomy prediction. Computational approaches, especially deep learning algorithms, are highly efficient for better pattern recognition and classification of large datasets compared to in silico techniques such as BLAST and machine learning methods. Here in this study, we present CNN_FunBar, a convolutional neural network-based approach for the classification of fungi ITS sequences from UNITE+INSDC reference datasets. Effects of convolution kernel size, filter numbers, *k*-mer size, degree of diversity and category-wise frequency of ITS sequences on classification performances of CNN models have been assessed at all taxonomic levels (species, genus, family, order, class and phylum). It is observed that CNN models can produce >93% average accuracy for classifying ITS sequences from balanced datasets with 500 sequences per category and 6-mer frequency features at all levels. The comparative study has revealed that CNN_FunBar can outperform machine learning-based algorithms (SVM, KNN, Naïve-Bayes and Random Forest) as well as existing fungal taxonomy prediction software (funbarRF, Mothur, RDP Classifier and SINTAX). The present study will be helpful for fungal taxonomy classification using large metagenomic datasets.

## 1. Introduction

Microorganisms are an inevitable part of the ecosystem. They play significant roles in organic matter decomposition [1,2], nutrient mineralization [3,4], atmospheric nitrogen fixation [5,6], bioremediation [7,8,9,10], etc. However, various harmful microbes are also present, which cause diseases in plants, animals as well as human beings. Although millions of diverse microbes exist in the environment as a highly interconnected network [11,12], only 1% of them can be isolated, cultured and studied through traditional wet lab experiments. Due to rapid advancement in sequencing methods, “Metagenomics” has emerged as a highly beneficial tool for researchers to characterize microbial community composition based on sequence data obtained directly from any environmental sample such as the human gut, infected tissue, crop rhizosphere, soil, ocean, etc. [13,14,15,16]. Taxonomy identification is the most crucial step of the metagenomic analysis pipeline [17,18,19]. Amplicon sequencing, i.e., DNA metabarcoding, has been found to be a highly successful approach [20,21,22,23]. In this method, a particular genomic region called marker gene is amplified using suitable primer pairs and sequenced, which can identify distinct species depending on the barcode gap. These marker genes vary according to the type of organisms. The 16S rRNA gene is the widely accepted marker gene for profiling prokaryotic (e.g., bacteria, archaea) microbes [24,25]. For eukaryotic species identification, several rRNA genes like large ribosomal subunit (LSU), small ribosomal subunit (SSU), the largest subunit of DNA-directed RNA polymerase II (RPB1), internal transcribed spacer (ITS region), etc. have been found as effective molecular markers [26].

Fungi are the second most abundant eukaryotic microbial domain consisting of around 3.8 million species and playing diverse roles as symbionts [27,28], mutualists [29,30], decomposers [31,32], antibiotic producers [33,34] as well as pathogens [35,36]. This domain consists of nine major phyla, viz., *Ascomycota*, *Basidiomycota*, *Opisthosporidia, Chytridiomycota, Neocallimastigomycota, Blastocladiomycota, Zoopagomycota, Mucoromycota* and *Glomeromycota* [37] among which *Ascomycota* and *Basidiomycota* are the largest (~70 to 90% of total fungal species) [38] and second largest (~35,000 species) [39] phyla, respectively.

Various DNA markers such as mitochondrial cytochrome c oxidase subunit 1 (CO1), protein-coding genes, viz., the largest subunit of RNA polymerase II (RPB1), the second largest subunit of RNA polymerase II (RPB II), minichromosome maintenance protein (MCM7), 18S rRNA, internal transcribed spacer (ITS) region, D1/D2 region of large ribosomal subunit (LSU), etc. have been used for fungal species identification [40]. Among these, the ITS region consisting of ITS1 and ITS2 hypervariable regions separated by comparatively conserved 5.8S segment (Figure 1) is the universally accepted DNA marker due to high variability in sequence composition and length among species [40]. Protein-coding DNA markers are highly efficient for phylogenetic analysis and fungal species identification. However, these genes have lower PCR (Polymerase Chain Reaction) amplification success rates compared to the ITS region. Apart from some early diverging fungal lineages, viz., *Pezizomycotina*, *Rozella*, *Neocallimasgomycota****,*** etc. and cryptic fungi species, ITS region is capable of discriminating a broad range of closely related fungal species with a high PCR amplification success rate [40].

Various ITS reference databases have been developed like UNITE (https://unite.ut.ee/; accessed on 20 October 2022) [42], Warcup training dataset (accessed on 20 October 2022) [43], BOLD database (https://boldsystems.org/; accessed on 21 October 2022) [44] and ITS1 database of NCBI GenBank (https://www.ncbi.nlm.nih.gov/; accessed on 21 October 2022) [45]. UNITE is a web-based database of nuclear ribosomal ITS sequences of various eukaryotic microorganisms. It collects all eukaryotic ITS sequences from the INSDC database and clusters them based on various similarity thresholds into species-level Operational Taxonomic Units (OTUs). These OTUs are called Species Hypothesis (SH), and they are assigned distinct Digital Object Identifiers (DOIs) for facilitating precise identification and assembly [42]. If two or more ITS sequences are available for a particular SH, one of them is randomly chosen to represent that SH. Currently, the UNITE database contains >6.4 million ITS sequences belonging to >0.2 million fungal species. However, more than 90% of a wide range of fungal species are still unknown [40]. The lack of reference sequences poses a significant challenge in fungal species delineation. Fungal taxonomy assignment can be done by aligning the query sequence against reference sequences through BLAST [46]. However, it is time-consuming and inefficient to identify novel species as compared to alignment-free machine learning-based algorithms. In recent years, a few machine learning-based fungal ITS classifiers have been developed, such as the RDP classifier [43,47], Mycofier [45], Mothur [48], SINTAX [49] and funbarRF [50]. The RDP classifier’s taxonomy assignment is based on 8-mer frequency features using the Naïve-Bayes algorithm [47]. Another Naïve-Bayes classifier, Mycofier, can classify fungal ITS1 sequences using 5-mer features up to the genus level [45]. Both Mothur [48] and SINTAX [49] use 8-mer features for fungi classification by the *k*-nearest neighbor (KNN) and non-Bayesian approaches, respectively. Random forest algorithm has been implemented in funbarRF for fungal species classification based on *g*-spaced di-nucleotide frequency feature representation [50].

Although various machine-learning approaches have been developed, there is enough scope for further research. Performances of the RDP classifier and SINTAX were evaluated using the Warcup training dataset (8551 species from 1461 genera) [43,49]. Mycofier was developed using ITS1 datasets (1794 species from 510 genera) of NCBI GenBank [45]. funbarRF was trained using the BOLD database (3674 species from 777 genera), and its performance was evaluated using various simulated and real metagenome datasets [50]. However, UNITE dataset covering a broad range of fungal species has not been extensively explored to develop machine learning-based classification approaches. The recent version of UNITE database (Version 9.0; Dated: 16 October 2022) comprises around 6,441,764 fungal ITS reference sequences belonging to 290,922 UNITE fungal SH with DOIs at 1.5% threshold (https://unite.ut.ee/#main; accessed on 21 October 2022). Recently, deep learning has become an effective paradigm for the classification and clustering of big data [51,52]. Deep learning algorithms like CNN and DBN have been found to be highly efficient for bacterial taxonomic classification based on whole shotgun metagenome as well as 16S rRNA amplicon sequencing datasets as compared to machine learning methods [53].

In this present work, we have evaluated the effects of convolution kernel size, number of filters, different *k*-mer sizes, degree of diversities and varying category-wise data frequencies in each taxonomic level on the performance of the convolutional neural network-based algorithm, CNN_FunBar, for classifying fungi ITS sequences using UNITE+INSDC dataset. We have compared the classification performance of CNN with four machine learning-based classification algorithms, i.e., SVM, KNN, Naïve Bayes and Random Forest. A comparative analysis between CNN and four existing ITS classification software, i.e., funbarRF [50], Mothur [48], RDP Classifier [43] and SINTAX [49], has also been carried out.

## 2. Materials and Methods

### 2.1. Fungal Barcode Datasets

The recent release of the UNITE+INSDC dataset (Version: 9.0, Release Date: 16 October 2022) comprising 6,441,764 fungi ITS sequences belonging to 20 fungal phyla has been downloaded from https://unite.ut.ee/repository.php; accessed on 21 October 2022. From this dataset, 5,530,925 sequences (85.86%) belonging to the 2 most abundant phyla, viz., *Ascomycota* (41.95%) and *Basidiomycota* (43.91%), are considered for this study. In this dataset, class *Agaricomycetes* (44.98%), order *Agaricales* (18.19%), family *Russulaceae* (7.22%), genus *Russula* (5.90%) and species *Russula_sp*. (5.82%) are found to be the most abundant class, order, family, genus and species, respectively. Redundant sequences, as well as those containing ambiguous characters other than A, T, G and C, have been removed from this dataset. After these filtering steps, the remaining dataset consists of 4,504,529 sequences (Table 1).

Among these, several ITS sequences that do not have clearly defined identifiers for one or more taxonomic levels have been found. In the FASTA header of such sequences, those identifiers are suffixed with “*Incertae_sedis*” for that particular category. For example, the ITS sequence with GenBank ID *UDB01488607* does not have a clearly defined class identity. So, in the dataset, its FASTA header is represented as:


*
>UDB01488607|k__Fungi;p__Basidiomycota;c__Basidiomycota_cls_Incertae_sedis;o__Basidiomycota_ord_Incertae_sedis;f__Basidiomycota_fam_Incertae_sedis;g__Basidiomycota_gen_Incertae_sedis;s__Basidiomycota_sp|SH1250579.09FU
*


Likewise, sequences corresponding to undefined order, family and genus are annotated as “*_ord_Incertae_sedis*”, “*_fam_Incertae_sedis*” and “*_gen_Incertae_sedis*”, respectively. ITS sequences for which genus names are known but distinct species names are not assigned, species identifiers of such sequences are represented as [genus_name]_*sp*. If the higher taxonomic level of a sequence is undefined, then all lower taxonomic levels are also found to be unknown. In this study, the performances of all classifiers have been evaluated in 6 taxonomic levels, and separate training datasets have been prepared for each level. To avoid ambiguity due to uncertainly defined identifiers at any taxonomic level, we have filtered those sequences from our dataset for that level. For example, to prepare the class-level training dataset, ITS sequences having class identifiers ending with “*_cls_Incertae_sedis*” are removed. Similarly, for order, family, genus, and species level training datasets preparation, sequence identifiers suffixed with “*_ord_Incertae_sedis*”, “*_fam_Incertae_sedis*”, “*_gen_Incertae_sedis*” and “*_sp*” for order, family, genus and species levels, respectively are removed (Figure 2).

For each of these six taxonomic-level datasets, the corresponding taxon is considered as the “category”. We have observed that the category-wise frequencies of ITS sequences are not uniform in these datasets. For example, in the family-level dataset, some families have >1000 ITS sequences, whereas some have even <50 ITS sequences. Similar scenarios have been observed for class, order, genus and species-level datasets. For example, in the species-level dataset, 35,613 out of 38,365 species have <20 ITS sequences. Balanced datasets with higher class frequencies are preferable for training the deep learning classifiers for better pattern recognition and higher model accuracies. Therefore, separate balanced datasets have been prepared for each taxonomic level using the stratified sampling method. For instance, to prepare family level-balanced dataset with 100 ITS sequences per category, 100 ITS sequences are randomly sampled from the family-level filtered ITS sequence dataset where 453 families have been found as strata with the minimum number of ITS sequences >100. So, the final dataset contains 45,300 sequences. In the same way, balanced datasets have been prepared for other taxonomic levels (Figure 2). The effects of diversity degrees, as well as category-wise data frequencies, on the performance of the CNN model, have been explored in this study. Hence, separate collections of balanced datasets with varying (Figure 2) and invariant unique categories have been prepared for each taxonomic level. Detailed information about these datasets is summarized in Table 2 and Table 3.

As only two phyla (i.e., *Ascomycota* and *Basidiomycota*) are considered in the present work, datasets having variation in the unique category numbers cannot be prepared for the phylum level (Table 2). Therefore, in-house python scripts have been used for data filtering as well as sampling to create these datasets.

### 2.2. Feature Vector Generation

ITS sequences are strings of A, T, G and C letters. Machine learning and deep learning models cannot process these data as such. In previous studies, *k*-mer composition features have been considered for classifier development [45,47,48,49,50]. Here, we have considered *k*-mer frequencies for training and evaluation of all the classifiers. For a DNA sequence of length L, the total number of distinct feature descriptors for a particular value of *k* will be $4^{k}$ and the total number of extracted *k*-mers will be (L−k+1). In this present work, three different values for *k*, i.e., *k* = 4, 5 and 6, have been considered to evaluate the classification performances of the CNN model as well as other machine learning algorithms. The total numbers of distinct feature descriptors are 256, 1024 and 4096 in our training datasets corresponding to 4-mers, 5-mers and 6-mers, respectively. Each ITS sequence is represented as frequencies of each feature descriptor for a particular *k* value. Normalization of *k*-mer frequencies has been performed so that features in the dataset can be assumed to follow the normal distribution with zero mean and unit variance. Normalized features can be calculated as follows:(1)$$z_{ij}=\frac{x_{ij}-\mu_{i}}{\sigma_{i}}$$

Here, $z_{ij}$ = normalized value for *j*’th instance of *i*’th feature, $x_{ij}\;$= actual value for *j*’th instance of *i*’th feature, $\mu_{i}$ = mean of *i*’th feature and $\sigma_{i}$ = standard deviation of *i*’th feature for all $i\in 4^{k}$ and *j* = total number of ITS sequences in the dataset.

The functions oligonucleotideFrequency from R-package Biostrings [54] and StandardScaler from Python library Scikit-Learn [55] have been used to compute *k*-mer frequencies followed by the data normalization.

### 2.3. Supervised Classifiers

We have considered one deep learning model, CNN and four machine learning algorithms, i.e., SVM, KNN, Naïve-Bayes and Random Forest, in this study. Brief descriptions of each classifier are presented in this section. Finally, RDP Classifier [47] is demonstrated in short.

#### 2.3.1. Convolutional Neural Network (CNN)

It is a deep learning algorithm that can efficiently classify image data. CNN model is also found to be effective for text and DNA sequence classification problems. It consists of an input layer, an output layer and four types of hidden layers: convolution layer, activation layer, pooling layer and fully connected layer. Features are given as inputs to the model through the input layer, and these are sequentially passed through four hidden layers before the final classification or prediction results are obtained from the output layer. Convolution kernels transform the input features and extract useful information from them [56]. In the activation layer, non-linear activation functions, i.e., ReLU, Sigmoid, tanh, etc., are applied to convolved features to handle the non-linearity present in the data. ReLU [57] is the most useful activation function, and its value ranges between 0 and 1. It converts any negative input from the previous layer to 0 so that the corresponding neuron is not activated. The pooling layer reduces the dimension of the input feature space. Finally, the input features are flattened to be represented in the fully connected layer for classification. The number of nodes in the output layer is the same as the number of classes in the input dataset. In this study, a 1D-CNN architecture derived from the original LeNet-5 [58] having two convolution layers of varying kernel sizes and filter numbers, followed by a ReLU activation layer and max pooling layer of size 2×1 with valid padding have been used (Figure 3).

In the hidden layer of the fully connected layer, 500 nodes have been considered with the Softmax activation function [59]. For hyperparameter tuning, the Adam optimizer [60] and Categorical crossentropy [61] loss function has been used. For the training of CNN, 100 epochs and 20 batch sizes have been found to be optimum.

#### 2.3.2. Support Vector Machine (SVM)

SVM [62] is highly efficient in classifying the input dataset into two distinct classes using two parallel hyperplanes. If the distance between these two parallel hyperplanes increases, the classifier becomes more accurate. Multiclass problems are considered a set of several binary classification cases, and classification is done iteratively. Kernel functions like gaussian, radial basis function, tanh, etc., are used for the non-linear transformation of input features. In this study, SVM with RBF kernel function having two parameters as C = 1000 and gamma = 0.0001 has been used.

#### 2.3.3. K-Nearest Neighbor (KNN)

KNN [63,64] is a simple and non-parametric supervised machine learning classifier for multi-class data. It checks for the similarity between the input feature instance and available data points and classifies the new data into the suitable class. Although KNN is effective and robust for large and noisy training datasets, it is prone to overfitting due to the inability to determine the optimum number of neighbors (*K*). In this study, GridSearch method has been applied to identify the optimum value of *K* for different datasets at various taxonomic levels.

#### 2.3.4. Naïve-Bayes Method

It is a probabilistic supervised classifier suitable for text classification based on the Bayes theorem [65]. The classification probability of a data point in a particular class is defined as the posterior probability, which is calculated as:(2)$$P\left(c_{i}|x\right)=\frac{P\left(x|c_{i}\right)\times P\left(c_{i}\right)}{P\left(x\right)}$$

For all i=1, 2, …, C; C = Total number of classes. $P\left(c_{i}|x\right)$ is the posterior probability of input data x to be classified as a member of i’th class. $P\left(x|c_{i}\right)$ = likelihood of x to be classified in $c_{i}$ class. $P\left(c_{i}\right)$ and P(x) are prior probabilities for input data x and class $c_{i}$ respectively. The assigned class for that particular data point is the one for which this posterior probability is found to be the highest. In this study, Gaussian Naïve-Bayes function has been used.

#### 2.3.5. Random Forest (RF)

Random forest [66] is an ensemble machine-learning classifier. Initially, a large number of candidate trees are generated from the bootstrap replicate sample of the input training dataset, and final class is decided based on the majority voting method. In this study, we have considered 500 initial candidate trees and gini entropy function.

#### 2.3.6. RDP Classifier

RDP Classifier [47] uses the Naïve-Bayes algorithm for rapid taxonomy assignment to microbial rRNA sequences based on 8-mer frequency features. It is a Java-based classifier, and source codes are available at https://github.com/rdpstaff/classifier (accessed on 20 October 2022). Before classifying new datasets, the classifier is needed to be retrained using available reference datasets from https://sourceforge.net/projects/rdp-classifier/files/RDP_Classifier_TrainingData/ (accessed on 20 October 2022) or using reference training datasets as per the choice of the researcher.

### 2.4. Training and Evaluation

Each dataset described in Table 2 and Table 3 is initially split into training and test dataset using unique categories as strata in a 90:10 ratio. Models are trained with the training dataset in a stratified 10-fold cross-validation procedure. In each fold, 90% of the training dataset is used for training the model, and the remaining 10% is used for validation. Values of evaluation metrics have been computed by taking averages of all 10 folds for the validation dataset. Final model performances are evaluated using the test dataset (Figure 4).

For the comparative study between CNN and other machine learning models, datasets with 100 sequences per category for class, order, family, genus and species levels and 500 sequences per category for phylum level are taken under consideration (Table 2). Training of machine learning models is done in a similar procedure mentioned before. Performances of all supervised classifiers have been evaluated based on the following metrics [67,68]:(3)$$Accuracy=\frac{TP+TN}{TP+TN+FP+FN}$$
(4)$$Precision=\frac{TP}{TP+FP}$$
(5)$$Recall=\frac{TP}{TP+FN}$$
(6)$$F1\;Score=\frac{2\times (\mathrm{Precision}\times \mathrm{Recall})}{\mathrm{Precision}+\mathrm{Recall}}$$
(7)$$MCC\;Score=\frac{TP\times TN-FP\times FN}{\sqrt{\left(TP+FP\right)\left(TP+FN\right)\left(TN+FP\right)\left(TN+FN\right)}}$$
where *TP*, *FP*, *TN* and *FN* are the number of True Positive, False Positive, True Negative and False Negative predictions, respectively. In this study, balanced datasets have been used. Hence, all the above-mentioned performance matrices have been found to perform equally well. Furthermore, the Multi-Criteria Decision-Making (MCDM) approach [69] and TOPSIS (Technique for Order Preference by Similarity to Ideal Solution) [70] have been applied to rank all the learning algorithms based on the aforementioned five evaluation metrics. R package Topsis [71] has been used for this analysis.

### 2.5. Comparison of CNN with Existing Fungi ITS Classification Software

The classification performance of CNN has been compared with RDP Classifier based on datasets having 100 sequences per category at each taxonomic level (Table 2). For CNN, 6-mer features have been considered. RDP Classifier has been retrained with the training dataset (90%) and evaluated using the testing dataset (10%). 10-fold cross-validation has been carried out.

Comparison of CNN model with existing four fungal barcode classification software, namely, funbarRF, RDP Classifier, Mothur and SINTAX, have also been conducted. ITS dataset of 13,630 sequences corresponding to 1363 fungal species (10 sequences per class) from funbarRF paper [50] has been used here. Evaluation has been done based on accuracy as well as the species identification success rate (*SISR*) [50]. *SISR* is calculated as follows:(8)$$SISR=\frac{\sum_{h=1}^{H}n_{h}}{\sum_{h=1}^{H}N_{h}}$$

Here, h = total number of distinct species/classes in the dataset = 1, 2, …, H. $N_{h}$ is the total number of sequences belonging to hth class, and $n_{h}$ the number of sequences correctly classified into hth class.

### 2.6. Implementation Details

Training and validation of all classifiers have been performed in Python 3.8.7 using Keras [72] library with Tensorflow backend [73]. RDP Classifier is downloaded from https://github.com/rdpstaff/classifier (accessed on 21 October 2022) [47]. The dataset used for comparing CNN with existing software is available at http://cabgrid.res.in:8080/funbarrf/dataset/ (accessed on 17 November 2022) [50]. All computational analyses have been carried out in a high-performance computing cluster of ASHOKA bio-computing resources at ICAR-Indian Agricultural Statistics Research Institute, New Delhi, India, with the following configuration:CPU: 64-bit Intel(R)-Xeon(R), 2.9 GHz, 1 TB;RAM: 128 GB;OS: Linux RedHat.

## 3. Results and Discussion

### 3.1. Effect of Convolution Kernel Size and Filter Numbers on CNN Model Performance

Species-level balanced dataset with 4-mer frequency features having category-wise frequency 100 (Table 2) has been used to compare the performance of various CNN model architectures for varying convolution kernel sizes and filter numbers. The dataset contains 42,900 ITS sequences belonging to 429 unique fungal species. Initially, the CNN architecture (i.e., CNN-1 in Table 4), as described in previous studies [53,74], is applied to the dataset. Afterward, the kernel size has reduced in each of these two convolution layers from five to three, and the number of filters in each layer has increased. In the second convolution layer, the number of filters is kept as double of the first convolution layer. It is observed that the improvements in all the evaluation metrics scores (i.e., accuracy, precision, recall, *F*1 *score* and *MCC score)* for three CNN architectures (i.e., CNN-2, CNN-3 and CNN-4 in Table 4) are very less, i.e., only ~1–2% increase in performance metrics scores as compared to the CNN-1 model architecture (Table 4) proposed earlier [53,74]. It is possibly due to very high sequence similarity (~99%) among 429 species and a lesser amount of data per category. Another possible reason may be the consideration of a lower *k* value. For larger *k* values, such differences will be more prominent due to the availability of more training data per category with the increased number of feature descriptors.

However, the CNN-3 model with two convolution layers of kernel size = 3 and numbers of filters = 32 and 64 in the first and second convolution layers, respectively, have been observed to generate the best classification performances in terms of accuracy, precision, recall, *F*1 *score* and *MCC score* on the test dataset (Table 4). Further increase in filter numbers has resulted in reduced model performance. Hence, the result indicates that the CNN-3 model is the optimum architecture for fungi ITS classification. This architecture has been used for other comparative analyses.

### 3.2. Impact of Diversity and k-mer Sizes on CNN Model Performance

Table 2 shows that when the category-wise data frequency is the lowest, the number of unique categories is the highest in our balanced datasets for all taxonomic levels. However, an increase in the unique category numbers leads to an increase in diversity levels for all taxonomic levels. Therefore, these datasets have been used to train and evaluate the CNN model with 4-mer, 5-mer and 6-mer frequency features to study the impact of diversity degrees as well as *k*-mer sizes. Results are presented in Table 5.

The result has suggested that the highest model accuracy has been obtained at the phylum (100%) level, followed by class (98.88%), order (98.33%), family (98.21%), genus (97.83%) and species (93.28%) levels, respectively (Table 5; Figure 5). It is due to lesser inter-class sequence similarities in higher taxonomic levels (e.g., phylum, class) compared to lower taxonomic levels (e.g., genus, species). The highest average accuracies (Table 5) have been obtained for training datasets containing 500 data per category (i.e., S6, G4, F4, O2, C2 and P1) in each taxonomic level (Table 2), but diversities due to unique categories in these datasets are found to be the lowest among others. The result also indicates that CNN can efficiently classify fungi ITS sequences with >94% accuracy up to the genus level and >89% accuracy at the species level using 6-mer feature vectors even when the diversity level is very high. Still, the category-wise data frequency is very low (e.g., 50 sequences per category). A previous study [74] revealed that the CNN model (CNN-1 model architecture in Table 4) could classify 1035 yeast species based on 6-mer frequency features belonging to *Ascomycota* and *Basidiomycota* phyla with an average accuracy of 82.84%. This model was found to be the most efficient classifier as compared to the other three classifiers, i.e., Deep Belief Network (DBN) (avg. accuracy = 80.40%), RDP Classifier (avg. accuracy = 71.45%) and BLAST Classifier (avg. accuracy = 81.12%). However, it is observed in the present study that CNN (CNN-3 model architecture in Table 4) has produced >86% accuracy for classifying 2752 different fungal species (i.e., more than twice the number of species classified earlier) with only 20 ITS sequences per class (Table 5; Figure 5).

In four out of six taxonomic levels (species, genus, family and phylum), the highest classification accuracies have been obtained with 6-mer features. It is due to the fact that with higher values of *k*, the number of distinct feature descriptors is also increasing, resulting in better training of the model. In order and class level, 5-mer features have generated better results than 6-mers, but differences in accuracies are not significant. We have also found similar results for other evaluation metrics in order and class levels (Appendix A). The possible reason may be the sampling fluctuation that occurred during the dataset preparation step. Hence, it can be inferred that CNN performs best with hexanucleotide frequency features. However, it has the potential to produce high classification accuracy (e.g., 93.03%) with lower values of *k*, i.e., *k* = 4, if class numbers are less (e.g., 64) or category-wise data frequencies (e.g., 500 sequences per class) are high even at the species level (Table 5).

### 3.3. Classification Performance Analysis Based on Varying Class Frequencies

In the previous analysis, it has been found that CNN performs better for datasets with category-wise higher frequencies. But in those datasets, unique category numbers were not constant. Hence, a separate set of datasets (Table 3) with invariant category numbers has been used for each taxonomic level. In addition, the performances of CNN models have been evaluated based on 6-mer frequency features in all taxonomic levels (Table 6).

It is found from the above results that at higher taxonomic levels (i.e., genus, family, order, class and phylum), an increase in all the performance metrics scores has been observed with the increase in category-wise data frequencies from 100 to 500. At the species level, average scores of all the performance metrics initially decreased when the category-wise data frequency was increased from 100 to 250. The species level is the lowest among all the taxonomic levels, and inter-species sequence similarity is very high (~99%), which indicates that the degree of heterogeneity is much less in species-level datasets. This is probably because the CNN model could not correctly recognize true patterns from the input features when the category-wise data frequency is increased from 100 to 250. Hence the increase in performance metrics scores may not have been observed with the increase in the category-wise data frequencies at the species level compared to other higher taxonomic levels. However, an increase in all the performance scores was observed when the category-wise data frequency was increased from 250 to 500 in the case of species-level datasets (Table 6; Figure 6). Overall, it can be seen that the average values of performance metrics scores are the highest for the largest value of category-wise data frequencies at all levels (Table 6; Figure 6). Due to the presence of more data points per class, better training of model parameters has been possible while capturing the maximum variability present in training datasets. In addition, larger data points are also available in respective validation and test datasets, leading to better model evaluation.

### 3.4. Comparison among Classification Performances of CNN and Other Machine Learning Algorithms

The classification performance of CNN is compared with SVM, KNN, Naïve-Bayes and Random Forest models using datasets with 100 sequences per category for species to class levels and 500 sequences per category at the phylum level (Table 2) while retaining a large number of classes. Average accuracies produced by these five classifiers for varying *k* values are presented in Table 7.

Similar to the previous study [74], it is also found that CNN has outperformed all machine learning classifiers at most taxonomic levels, especially at the species level (Table 7; Figure 7). Performances of CNN and Random Forest are found to be at par for all *k*-mer sizes. It can be observed that all five classifiers have produced the best classification results with hexanucleotide features in the majority of levels (Table 7; Figure 7). Some exceptions have been found for KNN; it obtained ~1% more accuracy with 5-mer features at class and family levels (Table 7). CNN also performed better with 5-mer frequencies at the class level. The average accuracies of CNN models are >2% higher than SVM and KNN classifiers up to the class level with hexanucleotide frequency features. However, it can be noticed that at the species level, CNN has obtained 4% and 3% more accuracies than SVM and KNN, respectively, with tetra-nucleotide frequencies (Table 7). The difference between the accuracy values of CNN and the Naïve-Bayes algorithm is >10% at this level. The Naïve-Bayes classifier developed the lowest accuracies among all classifiers at all levels. However, at higher levels, these accuracy gaps are found to be narrowed down, and the other three machine learning-based classifiers, as well as CNN, have obtained >93% classification accuracies. So, the result suggests that the stability of accuracy is highest for CNN and Random Forest and lowest for the Naïve-Bayes method. Similar results have been observed for mean precision, recall, *F*1 *scores* and *MCC scores* (Appendix A). Instead of the Naïve-Bayes algorithm, CNN, SVM, KNN and Random Forest models are further ranked based on the Topsis analysis. It can be observed from the results (Table 8) that apart from the class level, CNN has been ranked first among all four classification algorithms, with the highest score obtained at the other five taxonomic levels. At the phylum level, the same ranks (rank = 1) have been obtained from CNN, KNN and Random Forest with the highest scores of 1.0000.

Therefore, it can be inferred from the above results that CNN is the most efficient algorithm for classifying fungi ITS sequences compared to machine learning-based supervised classifiers.

### 3.5. Comparison of CNN with Existing ITS Classification Software

The classification performances of the RDP Classifier (*k* = 8 mer) [47] and CNN (*k* = 6 mer) have been compared using datasets with 100 sequences per category (Table 2). Table 9 and Figure 8a shows that the accuracy of the CNN model is ~10% more than RDP Classifiers from species to class level. At the phylum level, CNN has obtained 4% more accuracy than RDP Classifier.

However, the training time of the CNN model was much higher than RDP Classifier. Apart from this time complexity, it can be inferred that CNN is more efficient than RDP classifier as it can produce higher accuracy scores with fewer features at all taxonomic levels. Moreover, with the availability of advanced computational facilities such as powerful processors and efficient memory allocation, the time complexity problem can be overcome.

Further, the test dataset of the funbarRF paper [50] has been used to evaluate and compare the performance of the CNN model with funbarRF [50], Mothur [48], RDP Classifier [43] and SINTAX [49] in terms of accuracy score and species identification success rates (SISR) (Table 10; Figure 8b). The scores of the performance matrices for the existing four software are obtained from the above-mentioned paper.

From the above result, it can be noticed that with 6-mer features, the CNN model could correctly predict 148 more species than the funbarRF tool resulting in a high SISR of 89.89% (Table 10; Figure 8b). The average accuracy of CNN is also 2% higher than funbarRF software for this dataset. The previous study [50] revealed that funbarRF is better software for fungi species classification than Mothur, RDP Classifier and SINTAX in terms of both accuracy scores and the number of correct predictions. Hence, the above comparative analysis suggests that the CNN model (i.e., CNN_FunBar) is the most efficient fungal taxonomic classifier compared to all the existing software.

## 4. Conclusions

In this study, impacts of category diversity, *k*-mer size, and category-wise frequencies of ITS sequences on fungal ITS sequence classification by CNN have been assessed based on various performance metrics. UNITE+INSDC dataset has been used to cover a broad range of fungal taxa, and some important data filtering steps have been applied to avoid ambiguities. It is found that CNN architecture with two convolution layers of kernel size 3 and filter numbers 32 and 64 in the first and second layers, respectively, is the optimum one. Our study revealed that CNN had produced the highest classification accuracies for datasets with the highest number of sequences per category based on 6-mer frequency features. However, high accuracies have been obtained even when the dataset contains a large number of unique categories comprising a small number of sequences in each of them. The proposed CNN model architecture has been found to classify more than twice the number of fungal species with ~4% higher classification accuracy as compared to the previously used CNN model, which was found to be the best classifier among CNN, DBN, RDP Classifier and BLAST Classifier [74]. Our model has produced 86.12% average accuracy in classifying 2752 species based on hexamer nucleotide frequency features. The classification accuracy at the species level can further be improved by removing closely related species, cryptic fungi species and species complexes, as ITS sequence is not the most efficient DNA marker for their identification [40,74]. A comparative study has revealed that CNN can outperform machine learning algorithms for classifying fungi ITS barcodes even at the species level. CNN_FunBar is also found to be the most efficient taxonomy classification method among all existing software. Currently, ITS sequences from <1% of 3.8 million fungal species are available in the public domain, and many of these sequences are of poor quality [74]. This study highlights the necessity of retraining the developed algorithm with the availability of authenticated reference barcode datasets having higher category-wise data frequencies for developing better taxonomic classification algorithms.

## Figures and Tables

**Figure 1 genes-14-00634-f001:**
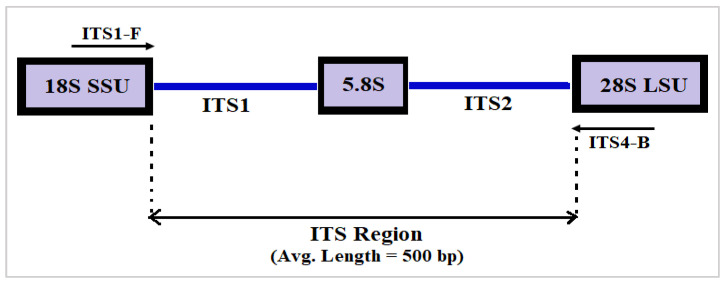
Schematic diagram of fungi ITS region consisting of ITS1 and ITS2 regions separated by 5.8S segment. ITS1-F and ITS4-B are widely used forward and reverse primers [41] to amplify the whole ITS region.

**Figure 2 genes-14-00634-f002:**
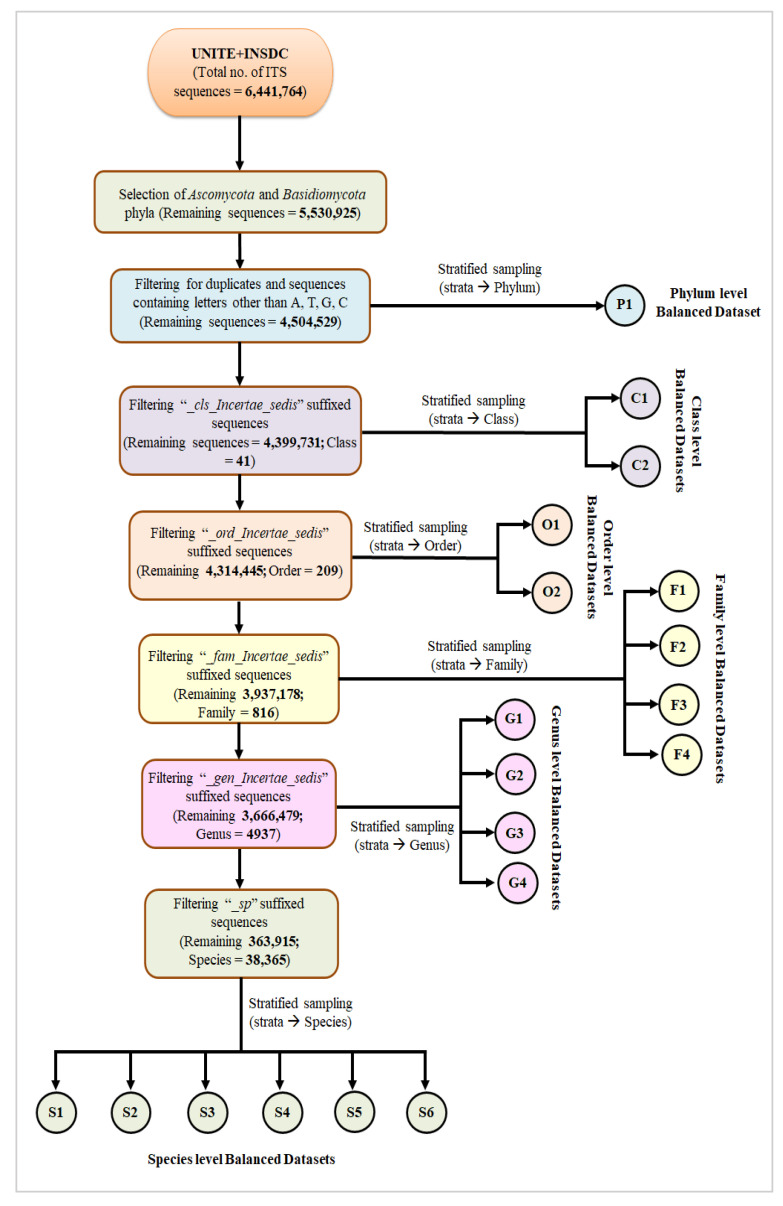
Diagrammatic representation of the pipeline for filtering and construction of balanced ITS sequence datasets at each taxonomic level.

**Figure 3 genes-14-00634-f003:**
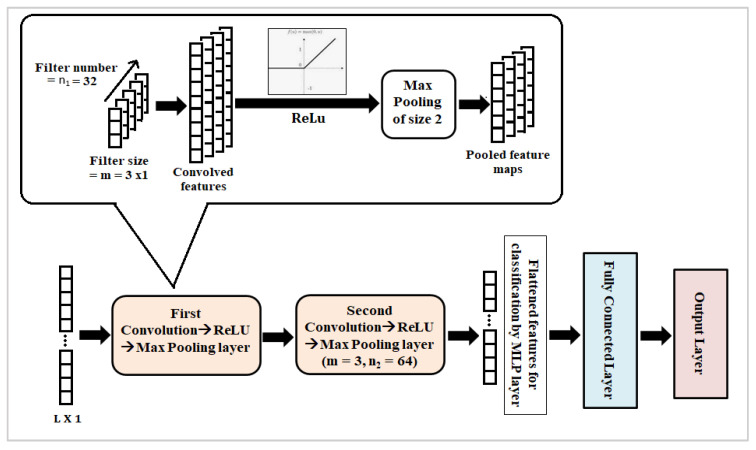
Architecture of the proposed CNN model. Here, m = size of convolution kernel, n_1_ = number of kernels in the first convolution layer and n_2_ = number of kernels in the second convolution layer.

**Figure 4 genes-14-00634-f004:**
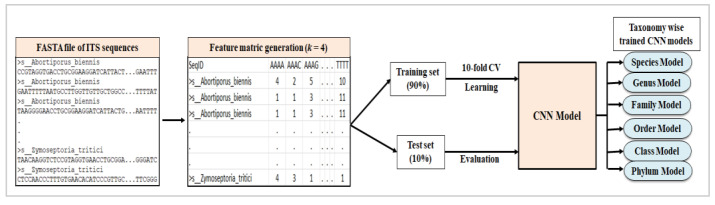
Pipeline for training and evaluation of CNN models starting from feature matrix generation from FASTA file of ITS sequences.

**Figure 5 genes-14-00634-f005:**
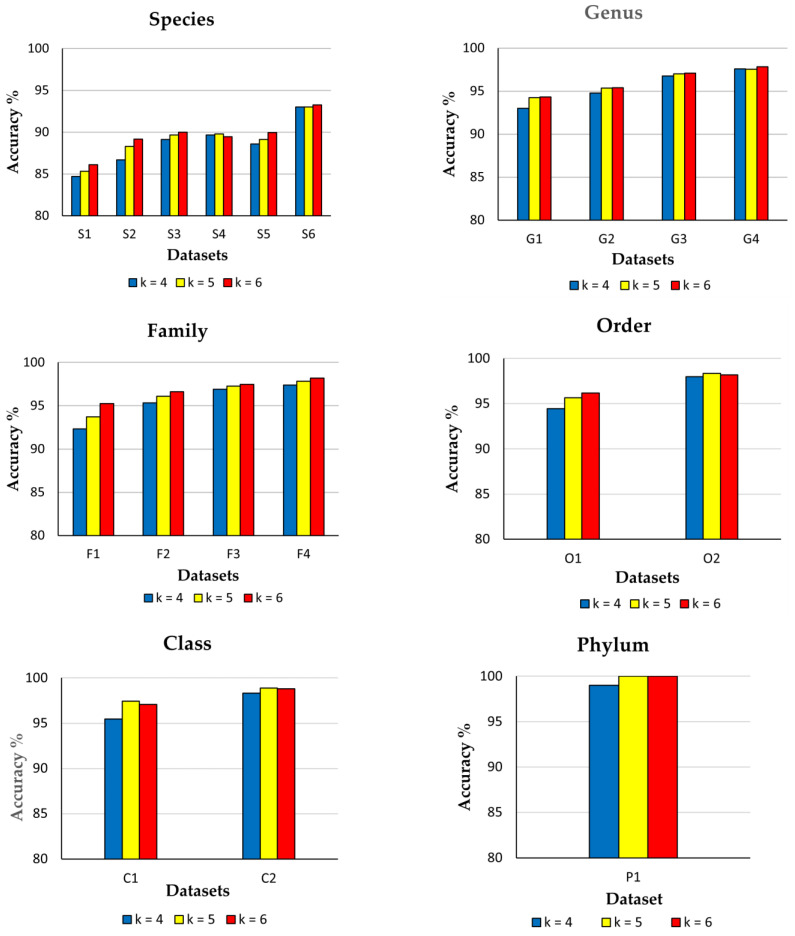
Average accuracy % of CNN for different datasets with varying *k*-mer sizes at all taxonomic levels.

**Figure 6 genes-14-00634-f006:**
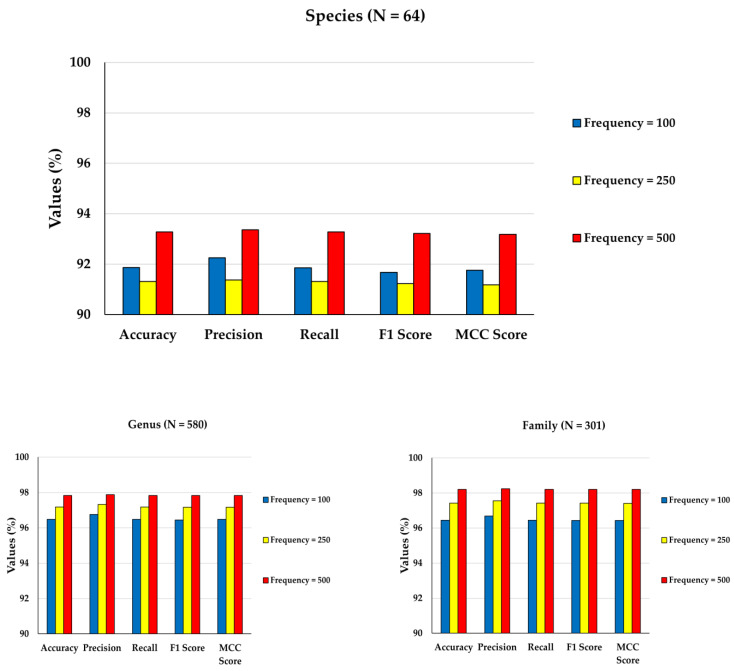
Average values (%) of evaluation metrics obtained from CNN models for balanced datasets at all taxonomic levels with invariant diversity levels and hexamer nucleotide frequency features. In each level, three datasets with 100, 250 and 500 data points per category are considered.

**Figure 7 genes-14-00634-f007:**
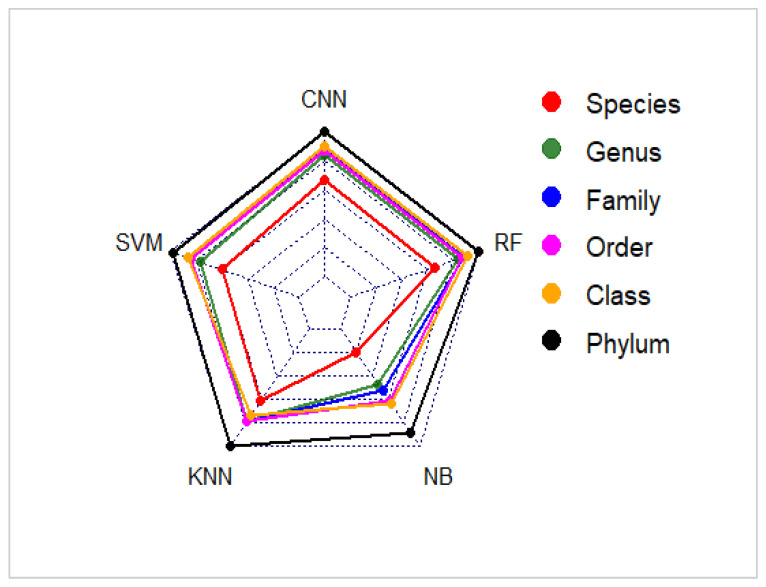
Average accuracies (%) obtained by CNN, SVM, KNN, Naïve-Bayes and Random Forest classifiers with 6-mer features at various taxonomic levels.

**Figure 8 genes-14-00634-f008:**
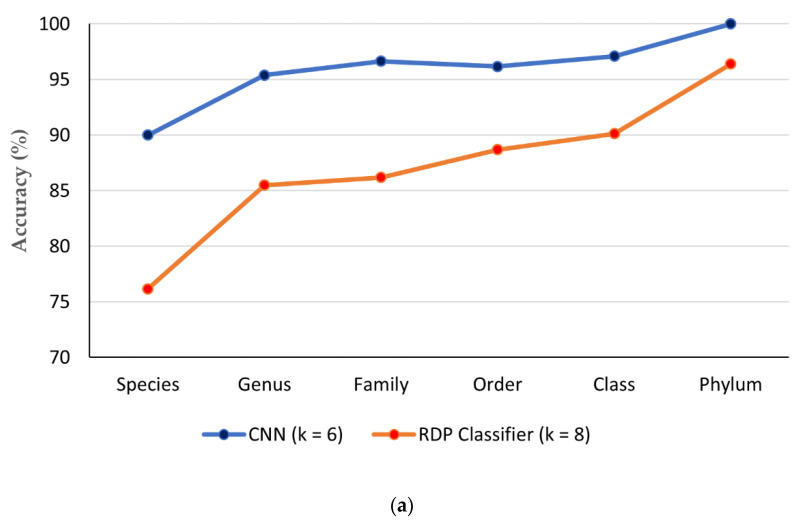
(**a**) Comparison between CNN and RDP Classifier based on average accuracy (%) at various taxonomic levels, (**b**) Comparison between CNN and existing software in terms of SISR (%) scores for funbarRF dataset [50].

**Table 1 genes-14-00634-t001:** Taxonomic level-wise distribution of ITS sequences in the dataset after filtering for duplicates and ambiguous nucleotides.

Phylum	Class	Order	Family	Genus	Species	Total ITS Sequences
2	49	274	1049	6367	44,167	4,504,529

**Table 2 genes-14-00634-t002:** Balanced datasets of ITS sequences with varying numbers of categories for all taxonomic levels.

Taxonomic Level	Datasets	No. of Unique Categories	No. of Sequence per Category	Total No. of ITS Sequences
Species	S1	2752	20	55,040
	S2	939	50	46,950
	S3	429	100	42,900
	S4	148	250	37,000
	S5	117	300	35,100
	S6	64	500	32,000
Genus	G1	1682	50	84,100
	G2	1293	100	129,300
	G3	841	250	210,250
	G4	580	500	290,000
Family	F1	520	50	26,000
	F2	453	100	45,300
	F3	369	250	92,250
	F4	301	500	150,500
Order	O1	149	100	14,900
	O2	106	500	53,000
Class	C1	31	100	3100
	C2	25	500	12,500
Phylum	P1	2	500	1000

**Table 3 genes-14-00634-t003:** Balanced datasets of ITS sequences with invariant category numbers for different taxonomic levels (from Species to Class levels).

Taxonomic Level	No. of Unique Categories	No. of Sequence per Category	Total No. of ITS Sequences
Species	64	100	6400
		250	16,000
		500	32,000
Genus	580	100	58,000
		250	145,000
		500	290,000
Family	301	100	30,100
		250	75,250
		500	150,500
Order	106	100	10,600
		250	26,500
		500	53,000
Class	25	100	2500
		250	6250
		500	12,500

**Table 4 genes-14-00634-t004:** Effect of kernel size and filter numbers on CNN model performance for classifying 100 sequences per species dataset with 4-mer features. Highest values are highlighted in bold.

CNN Model			Accuracy (%)	Precision (%)	Recall (%)	*F*1 *Score* (%)	*MCC Score* (%)
Name	Kernel Size (m)	No. of Filters in 1st and 2nd Convolution Layers (n_1_, n_2_)					
CNN-1	5	5, 10	87.88	87.74	87.86	87.84	87.85
CNN-2	3	16, 32	88.39	89.10	88.37	88.23	88.37
**CNN-3**	**3**	**32, 64**	**89.14**	**89.78**	**89.13**	**89.01**	**89.11**
CNN-4	3	64, 128	88.16	88.69	88.15	87.98	88.13

**Table 5 genes-14-00634-t005:** Average classification accuracy (%) of CNN model with varying *k*-mer sizes for balanced datasets at all taxonomic levels. Accuracy values obtained from the test dataset are presented in the table, and highest accuracies at each level are highlighted in bold.

Taxonomic Level	Datasets	Total No. of Levels	Accuracy (%) for Different *k*-mer Sizes		
			*k* = 4	*k* = 5	*k* = 6
Species	S1	2752	84.70	85.34	86.12
	S2	939	86.69	88.31	89.18
	S3	429	89.14	89.67	90.00
	S4	148	89.65	89.81	89.46
	S5	117	88.60	89.14	89.94
	S6	64	93.03	93.03	**93.28**
Genus	G1	1682	92.98	94.23	94.33
	G2	1293	94.78	95.37	95.39
	G3	841	96.74	97.00	97.10
	G4	580	97.59	97.54	**97.83**
Family	F1	520	92.35	93.73	95.23
	F2	453	95.32	96.11	96.64
	F3	369	96.89	97.25	97.49
	F4	301	97.37	97.82	**98.21**
Order	O1	149	94.43	95.64	96.17
	O2	106	98.00	**98.33**	98.19
Class	C1	31	95.48	97.42	97.09
	C2	25	98.32	**98.88**	98.80
Phylum	P1	2	99.00	100.00	**100.00**

**Table 6 genes-14-00634-t006:** Performance evaluation of CNN classifiers for datasets with unchanged diversity levels and hexanucleotide (*k* = 6) frequency features at various taxonomic levels. Highest values are highlighted in bold.

Taxonomic Level (N = No. of Unique Categories)	Sequence per Category	Average Values of Performance Metrics (%)				
		Accuracy	Precision	Recall	F1	*MCC*
Species (N = 64)	100	91.87	92.25	91.86	91.67	91.76
	250	91.31	91.37	91.31	91.23	91.18
	500	**93.28**	**93.36**	**93.28**	**93.22**	**93.18**
Genus (N = 580)	100	96.48	96.76	96.48	96.44	96.48
	250	97.18	97.32	97.18	97.17	97.17
	500	**97.83**	**97.88**	**97.83**	**97.83**	**97.83**
Family (N = 301)	100	96.44	96.69	96.44	96.43	96.43
	250	97.42	97.55	97.42	97.42	97.41
	500	**98.21**	**98.24**	**98.20**	**98.20**	**98.20**
Order (N = 106)	100	96.70	97.10	96.69	96.66	96.67
	250	97.32	97.48	97.32	97.32	97.30
	500	**98.19**	**98.25**	**98.19**	**98.19**	**98.17**
Class (N = 25)	100	97.60	97.84	97.60	97.63	97.51
	250	98.40	98.52	98.41	98.42	98.34
	500	**98.80**	**98.86**	**98.80**	**98.80**	**98.75**

**Table 7 genes-14-00634-t007:** Average accuracies (%) obtained by CNN, SVM, KNN, Naïve-Bayes and Random Forest classifiers with different *k*-mer sizes. For each taxonomic level (from Class to Species), the training dataset consists of 100 sequences per category. Phylum level dataset consists of 500 sequences per category. Highest values are highlighted in bold.

Taxonomic Level (N = No. of Unique Categories)	Value of *k*	Average Accuracy (%)				
		CNN	SVM	KNN	Naïve-Bayes	Random Forest
Species (N = 429)	4	89.14	85.8	86.71	74.68	89.12
	5	89.67	87.35	88.57	75.17	89.34
	6	**90.00**	**88.02**	**88.58**	**76.01**	**89.68**
Genus (N = 1293)	4	94.78	92.65	93.1	83.33	94.57
	5	95.37	92.98	93.52	84.00	95.01
	6	**95.39**	**93.07**	**93.64**	**84.12**	**95.17**
Family (N = 453)	4	95.32	94.13	93.07	85.29	94.56
	5	96.11	95.22	94.15	85.79	95.58
	6	**96.64**	**95.31**	**93.71**	**85.91**	**96.64**
Order (N = 149)	4	94.43	92.69	91.54	85.40	94.16
	5	95.64	94.02	93.69	87.00	95.63
	6	**96.17**	**95.27**	**93.73**	**88.54**	**96.04**
Class (N = 31)	4	95.48	93.62	92.52	88.38	95.45
	5	97.42	95.19	93.44	89.16	97.06
	6	**97.09**	**95.84**	**92.11**	**89.29**	**97.38**
Phylum (N = 2)	4	99.00	98.88	99.00	95.00	99.00
	5	100.00	99.55	99.01	96.00	99.00
	6	**100.00**	**99.55**	**100.00**	**96.52**	**100.00**

**Table 8 genes-14-00634-t008:** Ranking of CNN, SVM, KNN and Random Forest classifiers based on Topsis analysis at various taxonomic levels. Performances of all models have been evaluated based on five evaluation metrics (accuracy, precision, recall, *F*1 *score* and *MCC score*) with equal weightage using 6-mer frequency features. Highest ranks are highlighted in bold.

Taxonomic Level	Model	Score	Rank
Species	CNN	1.0000	**1**
	SVM	0.1701	4
	KNN	0.2335	3
	Random Forest	0.7877	2
Genus	CNN	0.9978	**1**
	SVM	0.1201	4
	KNN	0.2171	3
	Random Forest	0.9043	2
Family	CNN	1.0000	**1**
	SVM	0.6319	3
	KNN	0.2310	4
	Random Forest	0.9939	2
Order	CNN	1.0000	**1**
	SVM	0.6253	3
	KNN	0.2011	4
	Random Forest	0.9204	2
Class	CNN	0.9418	2
	SVM	0.6928	3
	KNN	0.3414	4
	Random Forest	1.0000	**1**
Phylum	CNN	1.0000	**1**
	SVM	0.7824	2
	KNN	1.0000	**1**
	Random Forest	1.0000	**1**

**Table 9 genes-14-00634-t009:** Average classification accuracy (%) of CNN and RDP Classifier.

Classifier	Taxonomic Levels					
	Species	Genus	Family	Order	Class	Phylum
CNN (*k* = 6)	90	95.39	96.64	96.17	97.09	100
RDP Classifier (*k* = 8)	76.18	85.50	86.20	88.69	90.13	96.41

**Table 10 genes-14-00634-t010:** Comparison among CNN model and other fungal ITS sequence classification software based on average accuracy (%) and SISR (%) values at the species level. Highest values of average accuracy (%), number of correct predictions and SISR (%) score are highlighted in bold.

Algorithm/ Software	Feature Type	Avg. Accuracy (%)	Correct Predictions	SISR (%)
	*k* = 4 NF	89.21	12,063	88.50
CNN	*k* = 5 NF	90.75	12,176	89.33
	*k* = 6 NF	91.27	12,252	89.89
FunbarRF (Random Forest)	*g*-spaced di-nucleotide features (*g* = 1 + 2 + 3 + 4 + 5)	~89	12,104	88.80
Mothur (KNN)	*k* = 8 NF	~89	12,062	88.49
RDP Classifier (Naïve-Bayes)	*k* = 8 NF	~87	11,864	87.04
SINTAX (Non-Bayesian)	*k* = 8 NF	~87	11,887	87.21

## Data Availability

The data used in this study are collected from the public domain, viz., https://unite.ut.ee/#main (accessed on 20 October 2022) and http://cabgrid.res.in:8080/funbarrf/dataset/ (accessed on 17 November 2022).

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
