# Peer review of "CNN_FunBar: Advanced Learning Technique for Fungi ITS Region Classification"

_genes, 2023, doi:10.3390/genes14030634_

Round 1

Reviewer 1 Report

Paper is ok and have great scientific contribution.

There are few suggestions

In the introduction section, authors used full form for LSU and SSU, so other regions like RPB1, and ITS should also be mentioned in full form

Also it would be great, if authors mention which order, family, genus and species have more sequences entry and which have minimum entry, it would be easy to predict which particular group is exploring more in the world

Author Response

We are thankful to the learned reviewer who has devoted so much time and his suggestions has definitely improved the quality of the paper. We have taken the reviewer’s comments seriously and accordingly the manuscript has been updated. The manuscript has been revised in the track-change mode and line numbers where the changes have been done are mentioned in the reply corresponding to each suggestion. Here we present one-by-one response to the issues raised by the reviewer.

Reviewer’s Comments and Suggestions for Authors:

Paper is ok and have great scientific contribution. There are few suggestions.

Reply: We are very grateful to the learned reviewer for duly acknowledging our efforts to carry out the present research work.

In the introduction section, authors used full form for LSU and SSU, so other regions like RPB1, and ITS should also be mentioned in full form.

  • In the introduction section, authors used full form for LSU and SSU, so other regions like RPB1, and ITS should also be mentioned in full form.

Reply: Full forms of other regions are incorporated in the revised manuscript (Line 46 and 47).

  • Also it would be great, if authors mention which order, family, genus and species have more sequences entry and which have minimum entry, it would be easy to predict which particular group is exploring more in the world.

Reply: We are thankful to the learned reviewer for this valuable remark. In the Materials and Methods section (Line 130-133), names of the most abundant class, order, family, genus and species along with their corresponding percentage values with respect to the total ITS sequences present in our filtered dataset are incorporated in the revised manuscript.

Reviewer 2 Report

The paper of Ritwika Das, Anil Rai, and Dwijesh Chandra Mishra entitled “CNN_FunBar: Advanced learning technique for fungi ITS region classification” is quite nice and well written. The CNN methodology could be better explained to be understandable by a wide audience. I found a few minor issues to be fixed in the text and have a few additional remarks or questions that need to be addressed.

Line 90, replace “UNITE database (Version 9.0; Dated: 17.10.2022)” with “UNITE database (Version 9.0; Dated: 16.10.2022)”

Line 216, replace “500 nodes are been considered” with “500 nodes have been considered”

Line 222, missing words? “If the distance between these hyperplanes is more, …”

Line 244, replace “In this study, Gaussian Naïve-Bayes function has been used here.” with “In this study, Gaussian Naïve-Bayes function has been used.”

Line 322, something missing? “The improvements in evaluation metrics scores are very less, i.e., …”

Line 331, unclear “In Table 2, it can be observed that category-wise data frequencies are needed to be kept  less  for  retaining  a  large  number  of  unique  classes  in  our  balanced  datasets.”

In Fig 6, how do you explain the lower percentages for Frequency 250 at species level compared to F100 and F500 while at the other taxonomic levels, the is a more logic gradation of the values between F100, F250 and F500? To me the following sentence is not in agreement with species level data: “It can be seen that average values of performance metrics are the highest for the largest value of category-wise data frequencies at all levels (Table 6; Figure 6)”.

As it seems that increase of the value of K of the kmer is almost always improving the global quality of the identifications at most levels, I would suggest testing what is the best K value given the tested datasets.

What I’m strongly missing in this study are both the times needed per sequence to classify them (for each algorithm, dataset and settings) and the amount of computer resources needed (CPU, GPU(?) and memory). These values are critically important as the number of sequences obtained from metagenomic studies can be very high and scalability can become a serious issue in the future.

Also, we all know that most species have not yet been sequenced or are known by a very little number of sequences. Obviously, the presented algorithms/software are very sensitive to these problems. I would like the authors to address these issues and determine which systems are better at working with such partial/incomplete datasets problems.

Author Response

We are thankful to the learned reviewer who has devoted so much time and his suggestions has definitely improved the quality of the paper. We have taken the reviewer’s comments seriously and accordingly the manuscript has been updated. The manuscript has been revised in the track-change mode and line numbers where the changes have been done are mentioned in the reply corresponding to each suggestion. Here we present one-by-one response to the issues raised by the reviewer.

Reviewer’s Comments and Suggestions for Authors:

The paper of Ritwika Das, Anil Rai, and Dwijesh Chandra Mishra entitled “CNN_FunBar: Advanced learning technique for fungi ITS region classification” is quite nice and well written. The CNN methodology could be better explained to be understandable by a wide audience. I found a few minor issues to be fixed in the text and have a few additional remarks or questions that need to be addressed.

Reply: We are very much thankful to the learned reviewer for appreciating our efforts to carry out the present research work. As per the given suggestion, we have incorporated more information about the CNN methodology in the Materials and Methods section (Line 223-227) in the revised manuscript.

  • Line 90, replace “UNITE database (Version 9.0; Dated: 17.10.2022)” with “UNITE database (Version 9.0; Dated: 16.10.2022)”.

Reply: The suggested correction has been incorporated in the Line 108 of the revised manuscript.

  • Line 216, replace “500 nodes are been considered” with “500 nodes have been considered”

 Reply: The suggested correction has been incorporated in the Line 241 of the revised manuscript.

  • Line 222, missing words? “If the distance between these hyperplanes is more, …”

Reply: The manuscript has been carefully revised and the suggested correction has been incorporated in the Line 247. The statement has been rewritten as “If the distance between these two parallel hyperplanes is more, the classifier becomes more accurate”.

  • Line 244, replace “In this study, Gaussian Naïve-Bayes function has been used here.” with “In this study, Gaussian Naïve-Bayes function has been used.”

Reply: The suggested correction has been incorporated in the Line 270 of the revised manuscript.

  • Line 322, something missing? “The improvements in evaluation metrics scores are very less, i.e., …”

Reply: As per the suggestion mentioned by the learned reviewer, the manuscript has been revised and the previous statement has been rewritten in Line 343-347.

  • Line 331, unclear “In Table 2, it can be observed that category-wise data frequencies are needed to be kept less  for  retaining  a  large  number  of  unique  classes  in  our  balanced  ”

 Reply: The suggestion has been carefully considered and the statement has been rewritten in Line 372-374 of the revised manuscript.

  • In Fig 6, how do you explain the lower percentages for Frequency 250 at species level compared to F100 and F500 while at the other taxonomic levels, the is a more logic gradation of the values between F100, F250 and F500? To me the following sentence is not in agreement with species level data: “It can be seen that average values of performance metrics are the highest for the largest value of category-wise data frequencies at all levels (Table 6; Figure 6)”.

Reply: We are thankful to the learned reviewer for pointing out this important issue which was overlooked by us previously. We have revisited the manuscript and have incorporated the suggested correction along with the probable justification in Line 434-445 of the revised manuscript. The species level is the lowest among all the taxonomic levels and inter-species sequence similarity is very high (~99%) in our dataset which indicates that the degree of heterogeneity is very less in species level datasets. Probably due to this fact, the CNN model could not correctly recognize true patterns from the features when the category wise data frequency is increased from 100 to 250 as compared to other higher taxonomic levels. Hence the gradual increase in performance metrics scores may have not been observed with the increase in the category wise data frequencies at species level.

  • As it seems that increase of the value of K of the kmer is almost always improving the global quality of the identifications at most levels, I would suggest testing what is the best K value given the tested datasets.

Reply: We appreciate the learned reviewer for this valuable remark. In our study, we have observed an increase in the classification performances of CNN models with the increase in kmer values from 4 to 6. Due to the time-constraint, we could not examine the classification results for higher k values like k = 7, 8, 9 etc. in this current study. We apologise to the learned reviewer for not being able to identify the optimum k value for our datasets in this current study. However, we consider this suggestion and shall carry out the analysis to identify the optimum k value as a part of our future research work.

  • What I’m strongly missing in this study are both the times needed per sequence to classify them (for each algorithm, dataset and settings) and the amount of computer resources needed (CPU, GPU(?) and memory). These values are critically important as the number of sequences obtained from metagenomic studies can be very high and scalability can become a serious issue in the future.

Reply: As per the suggestion, we have incorporated the information about the computer resources used in this study in Line 329-332 of Materials and Methods section under the subsection of Implementation Details. However, we did not conduct these analyses in a systematic manner keeping number of nodes same for all datasets and algorithms. Hence, we did not keep a record of training and classification times taken by each algorithm, setting and for various datasets. We apologise to the learned reviewer for not being able to mention time requirements of all the models in this current revised manuscript. However, we consider this suggestion and shall keep these records for documentation of further studies in future.

  • Also, we all know that most species have not yet been sequenced or are known by a very little number of sequences. Obviously, the presented algorithms/software are very sensitive to these problems. I would like the authors to address these issues and determine which systems are better at working with such partial/incomplete datasets problems.

Reply: The learned reviewer has very well pointed out the fact that most of the fungi species and still unknown and are needed to be sequenced. We have addressed the suggested issue in the Conclusions section of our revised manuscript (Line 562-565) which will be the main direction of our study in future. However, with the availability of more reference sequences in future, the developed algorithm can be retrained to capture new information from newly sequenced fungi species which may give better generalization power of the developed methodology.

Reviewer 3 Report

In this manuscript (ID genes-2103396), the author's effort was to improve and speed up the taxonomic assignments for the fungal identification for metagenomics samples, which has a significance to the scientific community.

The authors, however, should consider adequately adding more information on ITS. They assert (e.g., lines: 8-9) that “Internal Transcribed Spacer (ITS) region is the most efficient DNA marker for fungi taxonomy prediction“. Is it the best biomarker for fungal identification at the species level? Is ITS sufficient for fungal species delineating when there are closely-related species or species complexes?  

Moreover, the authors should consider adding more information on the necessity for publicly-available authenticated reference sequences as background information in the revised manuscript.

Finally, the authors should consider the results in context of findings by previous studies on the deep learning models, including those carried out in microbial taxonomic classification of metagenomic data. For instance, recent studies showed that CNN outperformed the traditional BLAST classification. It should be also for the Discussion section. It will be easier for the readers to follow the entire story.

Author Response

We are thankful to the learned reviewer who has devoted so much time and his suggestions has definitely improved the quality of the paper. We have taken the reviewer’s comments seriously and accordingly the manuscript has been updated. The manuscript has been revised in the track-change mode and line numbers where the changes have been done are mentioned in the reply corresponding to each suggestion. Here we present one-by-one response to the issues raised by the reviewer.

Reviewer’s Comments and Suggestions for Authors:

In this manuscript (ID genes-2103396), the author's effort was to improve and speed up the taxonomic assignments for the fungal identification for metagenomics samples, which has a significance to the scientific community.

Reply: We are thankful to the learned reviewer for duly acknowledging our efforts to carry out this present study.

  • The authors, however, should consider adequately adding more information on ITS. They assert (e.g., lines: 8-9) that “Internal Transcribed Spacer (ITS) region is the most efficient DNA marker for fungi taxonomy prediction“. Is it the best biomarker for fungal identification at the species level? Is ITS sufficient for fungal species delineating when there are closely-related species or species complexes?

Reply: As per the suggestion, the phrase “the most efficient” has been replaced with “a potential” in the Line 9 of the revised manuscript. We have also mentioned the names of other DNA markers applicable for fungal species prediction in Line 57-61. We have also mentioned the advantages of ITS region over other markers in the context of fungal species delineation in Line 64-70. We are thankful to the learned reviewer for pointing out the fact that ITS sequence may not be the most efficient biomarker for discriminating closely-related fungal species or species complexes. Therefore, the possible approach to overcome this issue has been mentioned in Line 556-559 of the revised manuscript.

  • Moreover, the authors should consider adding more information on the necessity for publicly-available authenticated reference sequences as background information in the revised manuscript.

Reply: As per the suggestion given by the learned reviewer, the necessity for publicly-available authenticated reference sequences has been mentioned in the Background information (Line 85-88) as well as in the Conclusions section (Line 562-565).

  • Finally, the authors should consider the results in context of findings by previous studies on the deep learning models, including those carried out in microbial taxonomic classification of metagenomic data. For instance, recent studies showed that CNN outperformed the traditional BLAST classification. It should be also for the Discussion section. It will be easier for the readers to follow the entire story.

 Reply: We are thankful to the learned reviewer for this valuable remark. We have incorporated this suggestion in the Results and Discussion section (Line 393-401) as well as in the Conclusion section (Line 552-556) of the revised manuscript.

Reviewer 4 Report

Major Comments

In general, the manuscript is interesting about the computational approaches for fungal taxonomy classification in large metagenomics datasets. However, some minor corrections could be needed in the current version of this manuscript in order to provide a better compression of the same. In addition, the authors should revise some comments by sections of the manuscript to improve it, as follow:

General

In several lines (144,202,213,221,281,289,312,315,355,406) in the text the authors used numbers or write the number, they should be more consistent, in general when the number is below that ten should be written.

Introduction

Lines 32-33 what are the authors mean by “in traditional wet lab conditions”? could be better to use another example for this sentence.

In line 44, SSU and the 18S region are the same so delete one of them.

In lines 49-50, the reference used here should be actualised, due to the phylum Zygomycota does not exist anymore and are more than five phyla too.

In line 67, the authors did not provide OTUs definition and DOIs either.

Materials & Methods

Line 170, the word is variation or invaration?

Results

Line 345, the phrase “datasets containing 500 data per category” this information does not appear in table 5.

Author Response

We are thankful to the learned reviewer who has devoted so much time and his suggestions has definitely improved the quality of the paper. We have taken the reviewer’s comments seriously and accordingly the manuscript has been updated. The manuscript has been revised in the track-change mode and line numbers where the changes have been done are mentioned in the reply corresponding to each suggestion. Here we present one-by-one response to the issues raised by the learned reviewer.

Reviewer’s Comments and Suggestions for Authors:

Major Comments

In general, the manuscript is interesting about the computational approaches for fungal taxonomy classification in large metagenomics datasets. However, some minor corrections could be needed in the current version of this manuscript in order to provide a better compression of the same. In addition, the authors should revise some comments by sections of the manuscript to improve it, as follow:

Reply: We are very grateful to the learned reviewer for duly acknowledging our efforts to carry out this present research work.

General

  • In several lines (144,202,213,221,281,289,312,315,355,406) in the text the authors used numbers or write the number, they should be more consistent, in general when the number is below that ten should be written.

Reply: As per the suggestion given by the learned reviewer, corrections have been incorporated in the revised manuscript (Lines: 165, 223, 238, 246, 306, 314, 340, 364, 405 and 470).

Introduction

  • Lines 32-33 what are the authors mean by “in traditional wet lab conditions”? could be better to use another example for this sentence.

Reply: The sentence has been rewritten by replacing the phrase “in traditional wet lab conditions” with “through traditional wet lab experiments” in Line 33-34 of the revised manuscript.

  • In line 44, SSU and the 18S region are the same so delete one of them.

 Reply: As per the suggestion, 18S rRNA is deleted and only SSU is kept in the revised manuscript (Line 46).

  • In lines 49-50, the reference used here should be actualised, due to the phylum Zygomycota does not exist anymore and are more than five phyla too.

Reply: We are thankful to the learned reviewer for pointing out this major mistake in the submitted manuscript. We have incorporated the current information, i.e., currently there exists nine major fungal phyla…, in the revised manuscript (Line 52-54). We have also mentioned the correct reference of this information in the Reference section (Line 680-681).

  • In line 67, the authors did not provide OTUs definition and DOIs either.

Reply: Full forms of both OTU (Line 81) and DOI (Line 82) are mentioned in the revised manuscript.

Materials & Methods

  • Line 170, the word is variation or invaration?

Reply: The sentence has been rewritten in the revised manuscript (Line 190-192) and the Table number is also added to clarify the meaning of the statement. As we have considered only 2 phyla, therefore we could not prepare separate balanced datasets having variation in the unique category numbers at phylum level. So, we find that the word “variation” is justified in this context and have kept unchanged in the revised manuscript.

Results

  • Line 345, the phrase “datasets containing 500 data per category” this information does not appear in table 5.

Reply: As per the suggestion of the learned reviewer, we have rechecked the statement and have rewritten it by mentioning proper reference table numbers (Line 387-389) in the revised manuscript.

Round 2

Reviewer 2 Report

The authors have properly responded even if I would have loved to see the results of speed and kmer number increase tests. Hope they will include them in further studies

Reviewer 3 Report

The authors have improved the manuscript. The revised version should be considered for publication.